# Transcriptome Dynamics of *Pseudomonas aeruginosa* during Transition from Overlapping To Non-Overlapping Cell Cycles

Kathrin Alpers,[a] Elisabeth Vatareck,[a] Lothar Gröbe,[b] Mathias Müsken,[c] Maren Scharfe,[d] Susanne Häussler,[a,e,f,g] Jürgen Tomasch[a,h]

[a]Department of Molecular Bacteriology, Helmholtz Centre for Infection Research, Braunschweig, Germany

[b]Platform Flow Cytometry and Cell Sorting, Department of Experimental Immunology, Helmholtz Centre for Infection Research, Braunschweig, Germany

[c]Central Facility for Microscopy, Helmholtz Centre for Infection Research, Braunschweig, Germany

[d]Platform Genome Analytics, Helmholtz Centre for Infection Research, Braunschweig, Germany

[e]Institute for Molecular Bacteriology, Twincore, Centre for Clinical and Experimental Infection Research, Hannover, Germany

[f]Department of Clinical Microbiology, Copenhagen University Hospital – Rigshospitalet, Copenhagen, Denmark

[g]Cluster of Excellence RESIST (EXC 2155), Hannover Medical School, Hannover, Germany

[h]Institute of Microbiology of the Czech Academy of Science, Center Algatech, Třeboň, Czech Republic

**ABSTRACT** Bacteria either duplicate their chromosome once per cell division or a new round of replication is initiated before the cells divide, thus cell cycles overlap. Here, we show that the opportunistic pathogen *Pseudomonas aeruginosa* switches from fast growth with overlapping cell cycles to sustained slow growth with only one replication round per cell division when cultivated under standard laboratory conditions. The transition was characterized by fast-paced, sequential changes in transcriptional activity along the *ori-ter* axis of the chromosome reflecting adaptation to the metabolic needs during both growth phases. Quorum sensing (QS) activity was highest at the onset of the slow growth phase with non-overlapping cell cycles. RNA sequencing of subpopulations of these cultures sorted based on their DNA content, revealed a strong gene dosage effect as well as specific expression patterns for replicating and nonreplicating cells. Expression of flagella and *mexE*, involved in multidrug efflux was restricted to cells that did not replicate, while those that did showed a high activity of the cell division locus and recombination genes. A possible role of QS in the formation of these subpopulations upon switching to non-overlapping cell cycles could be a subject of further research.

**IMPORTANCE** The coordination of gene expression with the cell cycle has so far been studied only in a few bacteria, the bottleneck being the need for synchronized cultures. Here, we determined replication-associated effects on transcription by comparing *Pseudomonas aeruginosa* cultures that differ in their growth mode and number of replicating chromosomes. We further show that cell cycle-specific gene regulation can be principally identified by RNA sequencing of subpopulations from cultures that replicate only once per cell division and that are sorted according to their DNA content. Our approach opens the possibility to study asynchronously growing bacteria from a wide phylogenetic range and thereby enhance our understanding of the evolution of cell cycle control on the transcriptional level.

**KEYWORDS** cell cycle, DNA replication, *Pseudomonas aeruginosa*, transcription

Bacteria differ in the ways replication is coordinated with cell growth and division (1). In fast-growing representatives, such as the model organisms *Escherichia coli* or *Bacillus subtilis*, a new round of replication is initiated before the previous one is finished. This overlap of cell cycles leads to multiple replication forks inside one cell and allows the speed of cell division to exceed the time required for chromosome duplication. The result is a strong gene dosage gradient along the origin (*ori*)-terminus (*ter*) axis of the chromosome. The higher gene copy number closer to *ori* can be exploited

Address correspondence to Susanne Häussler, Susanne.Haeussler@helmholtz-hzi.de, or Jürgen Tomasch, tomasch@alga.cz.

The authors declare no conflict of interest.

to maximize expression of traits needed during rapid growth and to control gene expression (2, 3). It has been shown that moving an *ori*-located *Vibrio cholerae* gene cluster coding for ribosomal proteins close to *ter* reduced the growth rate of the culture, while the wild-type growth level could be restored by placing two copies of this cluster at *ter* (4). Furthermore, the timing of spore formation in *B. subtilis* is an example for dosage imbalances triggering regulatory events between genes located on opposite ends of the replicating chromosomes (5). In slow-growing bacteria, the chromosome is duplicated only once per cell division (6). In several bacterial phyla, a differentiation program is triggered during these non-overlapping cell cycles. The best studied model is the bi-phasic lifestyle of *Caulobacter crescentus*. In this bacterium, a complex gene regulatory network precisely times the development of a flagellated from a stalked cell during replication and cell division (7).

*Pseudomonas aeruginosa* is a ubiquitous environmental bacterium, but also an opportunistic pathogen frequently causing nosocomial infections of various body sites, such as the lung, bloodstream, urinary tract, and burn wounds (8). Furthermore, *P. aeruginosa* poses a particular threat to patients suffering from cystic fibrosis (CF) (9). During livelong chronic infections of the CF lung, the bacterium adapts and evolves toward a slow growing phenotype (10). Doubling times are estimated to be around 30 min under laboratory conditions in lysogeny broth (LB) medium and 1.9 to 4.6 h in the CF lung (11). The cell cycle dynamics of *P. aeruginosa* has been extensively studied. Its chromosome is oriented with *ori* close to the center of the cell and *ter* located at the cell pole where the division plane forms. During replication, both *ori* move to the poles of the elongated predivisional cell where another round of replication can be started (12, 13). Despite the huge body of comparative transcriptome data available for this important pathogen (14–16), the effect of replication on gene expression has not explicitly been studied yet.

Here, we monitored growth and cell division and recorded a time-resolved transcriptome of *P. aeruginosa* PA14 in LB medium over 10 h at 1-h intervals. We show that the culture switches from fast growth with overlapping to sustained growth with non-overlapping cell cycles. The transition is characterized by fast-paced, sequential changes in transcriptional activity along the *ori-ter* axis. Furthermore, we identified replication- and nonreplication-associated gene expression during growth with non-overlapping cell cycles using a newly developed protocol based on fluorescence-activated cell sorting (FACS).

## RESULTS

**Growth and replication dynamics of *P. aeruginosa* in LB medium.** In accordance with previous reports (11), *P. aeruginosa* cultures reached an $OD_{600}$ of 1.8 $\pm$ 0.24 from a starting $OD_{600}$ of 0.05 within 4 h and a maximum doubling time of 34 $\pm$ 1 min when grown under standard laboratory conditions (Fig. 1A). This exponential growth phase was followed by slower growth to a maximum $OD_{600}$ of 3.17 $\pm$ 0.11 after 9 h with an OD value doubling time of 410 $\pm$ 110 min. Cell numbers, too, increased exponentially in the first 4 h from $4.5 \times 10^7$ to $7.3 \times 10^8 \pm 1.8 \times 10^8$ cells/mL with a doubling time of 30 $\pm$ 9 min, followed by decreased growth to a maximum count of $3.7 \times 10^9 \pm 5.8 \times 10^7$ cells/mL after 9 h with a doubling time of 168 $\pm$ 6 min. The notably slower increase of $OD_{600}$ values compared to cell numbers in the last 6 h of cultivation could be explained by a decrease in cell size at later growth stages that is indicative for reductive cell division (Fig. 1B; Fig. S1A).

The chromosome content of cells was monitored by stoichiometric staining with SYBR green (Fig. 1C; Fig. S1B). In the overnight grown precultures that were used for inoculation, 80% of the cells contained one chromosome (C1). One hour after the transfer into fresh medium, already 62% $\pm$ 7% and 30% $\pm$ 2% of the cells contained two (C2) and three chromosome equivalents, respectively, and a smaller fraction even more. This clearly indicates that the cultures had moved to a phase of growth with overlapping cell cycles. After 4 to 5 h of growth, the chromosome content shifted back

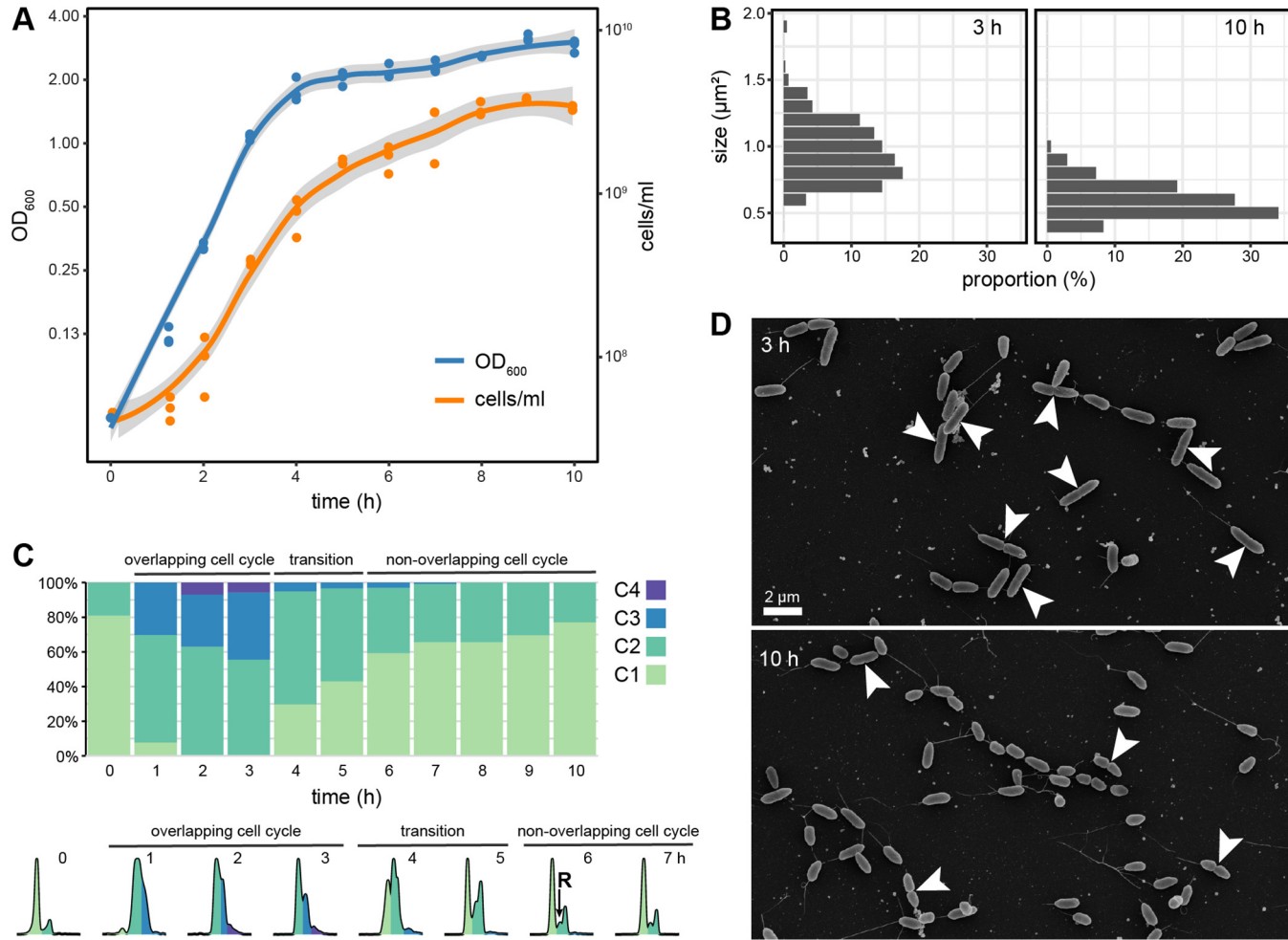

**FIG 1** Growth and Replication dynamics of *P. aeruginosa* in LB medium. (A) Optical density and cell numbers followed for 10 h of growth in LB medium. (B) Distribution of cell area as determined from EM micrographs. (C) Distribution of chromosome content revealed by flow cytometric analysis of SybrGreen fluorescence (One to four chromosome equivalents indicated by color). The lower panel shows representative distributions of fluorescence intensity for up to 7 h. R indicates replicating cells during growth with non-overlapping cell cycles. (D) Representative EM micrographs of cells during overlapping (3 h) and non-overlapping (10 h) cell cycles. Visible division planes are marked by a white arrow.

and two distinct peaks for cells containing one or two chromosomes became visible again. The strong C1 peak indicated that the majority of cell do not actively replicate at the time point of observation. However, the presence of cells with a DNA content between the major peaks indicated actively replicating cells (R). The proportion of C2 and R cells was only slowly reduced from 38% $\pm$ 3% to 23% $\pm$ 2% between 6 h and 10 h of growth. The presence of replicating cells after 10 h of cultivation was also visible on electron micrographs (Fig. 1D). Our data strongly suggest that *P. aeruginosa* shifts from fast growth with overlapping to sustained growth with non-overlapping cell cycles during the course of cultivation in LB medium with a short transition phase in between (Fig. 1C).

**Transcriptome dynamics of *P. aeruginosa* during different growth phases.** We monitored transcriptional changes for the full growth period in 1-h intervals. Two independent experiments with two and three replicates each were carried out. The transcriptomes clustered according to the growth phases except for the 6-h samples. For these samples, the transcriptomes of the first experiment were closer to the transition phase, while the transcriptomes of the second experiment were closer to the samples with non-overlapping cell cycles (Fig. S2A). The 1,736 genes, which showed a significant differential expression during the course of cultivation, could be assigned to eight clusters (Fig. S2B and C; Table S1).

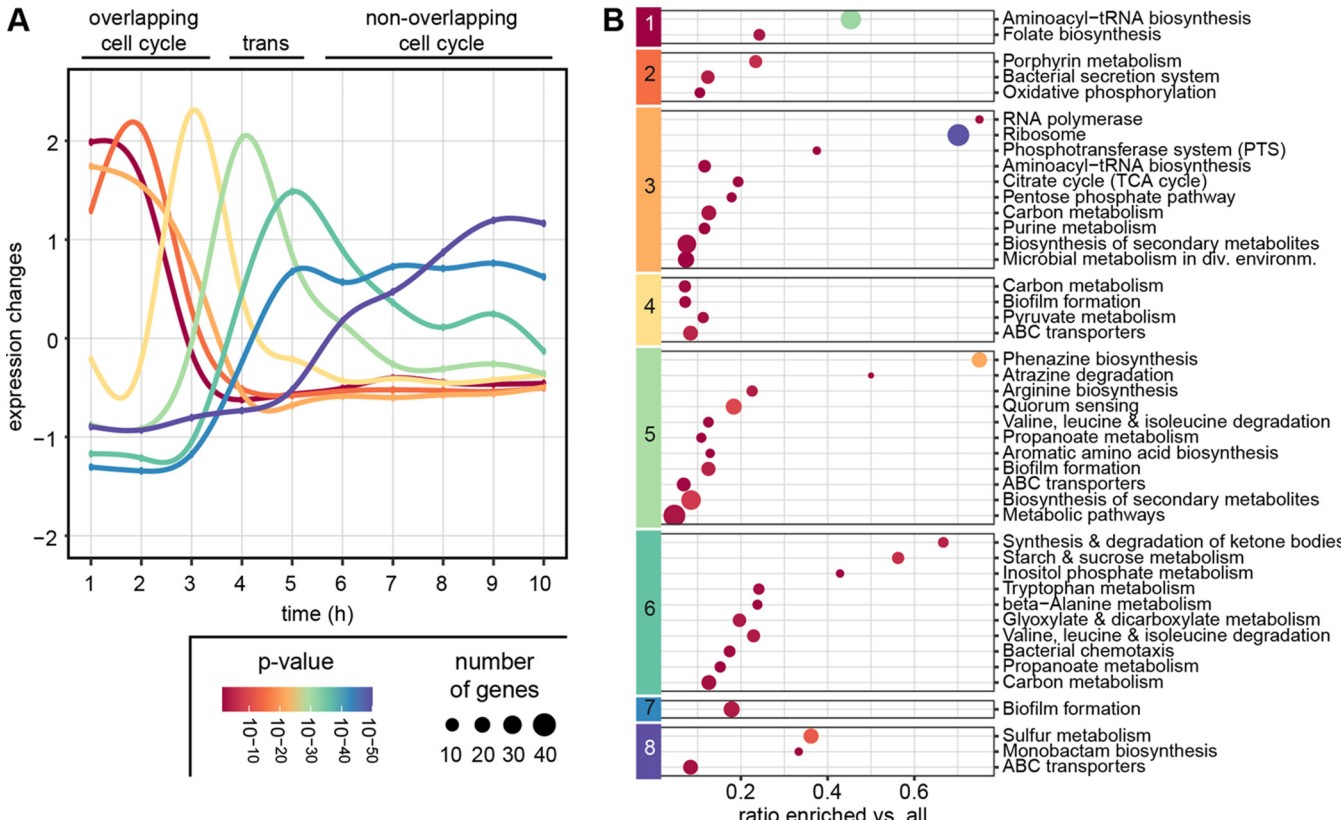

**FIG 2** Transcriptome dynamics during growth in LB medium. (A) Expression dynamics of the eight clusters determined with mfuzz. Shown are the changes of the average expression in the according clusters during the course of a 10 h-cultivation. (B) Significantly ($P < 0.05$) enriched KEGG-categories in the eight clusters. Size indicates the number of enriched genes in the category, color is according to $P$-value corrected for false discovery rate.

The transition between growth-phases was characterized by fast-paced waves of transient transcriptional activity (Fig. 2A). Genes in clusters 1 to 3 showed a comparable high expression during the first 2 h of growth, but with a different timing of maximum expression and the decline afterwards. During this growth phase, in particular transcription- and translation-related processes were expressed (Fig. 2B), including biosynthesis of tRNAs, RNA polymerase and ribosomes as well as chaperones. A high activity was seen for oxidative phosphorylation and also for biosynthesis of the vitamins folate (B9) and cobalamin (B12), in accordance with their respective roles in DNA and methionine synthesis. Expression of the type III secretion system gene clusters *psc* and *pcr* and the *exoT* effector (17) peaked at 2 h of cultivation followed by a steep decline.

Cluster 4 to 6 contained genes that showed a transient high expression at the end of exponential growth. The high number of sugar and amino acid transporters as well as genes of the pyruvate metabolism indicated a shift in metabolic preferences. In particular transporters for branched-chain amino acids were found to be active in this transition phase, in accordance with their late utilization as a carbon source observed before (10). Increased relative expression of the urea cycle and denitrification, and the glycogen metabolism pathway indicate changes of nitrogen and carbon utilization at this stage. Cluster 7 harbored genes, for which transcripts increased in abundance late in the transition phase and exhibited a stable expression throughout growth with non-overlapping cell cycles. Denitrification genes were among them as well as genes encoding subunits of a sulfate transporter and the MexHIG antibiotic efflux pump (18). Finally, relative expression of the late responding genes in cluster 8 increased between 5 and 8 h before reaching a stable level. In particular, high abundance of the pyoverdine biosynthesis machinery, the heme acquisition protein HasA and the sulfonate transport and metabolism pathway indicate a response to iron and sulfur limitation in the medium, respectively.

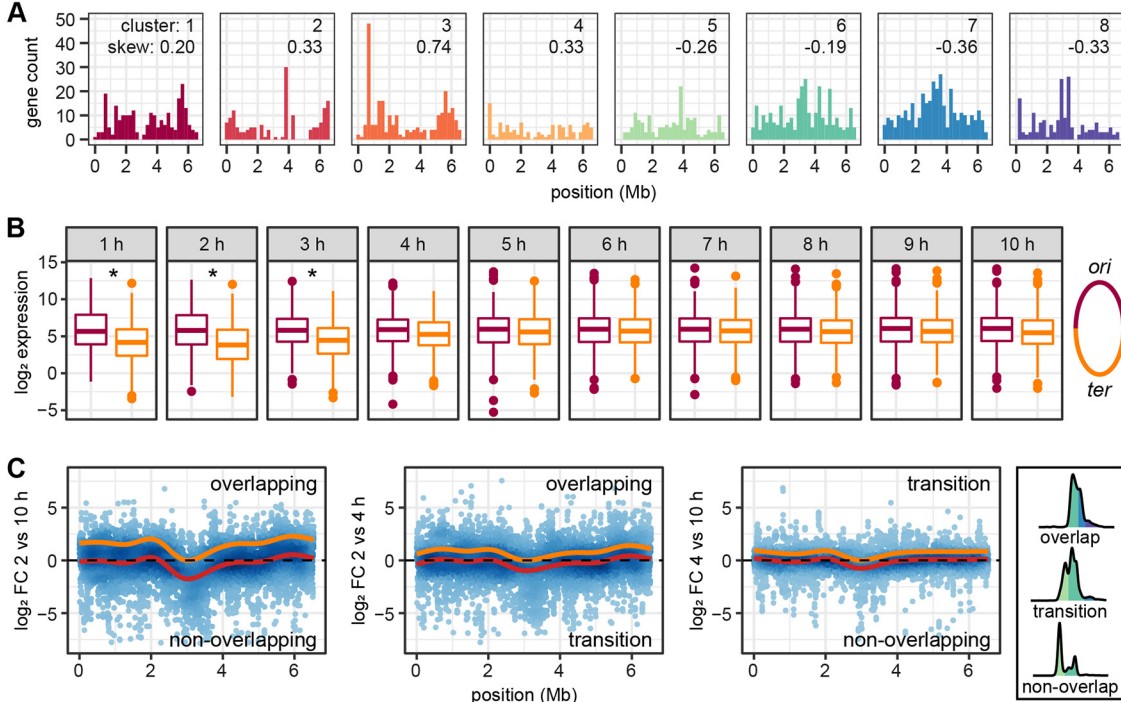

**FIG 3** Global chromosomal gene expression changes between different growth phases. (A) Distribution of genes on the chromosomes that show the highest relative expression during growth with overlapping cell cycles (clusters 1 to 3), transition phase (clusters 4 to 6), and growth with non-overlapping cell cycles (clusters 7 to 8, see Fig. 2A). Skew indicates the skewness of the distribution of positions relative to *ori* of the genes in these clusters. Right- and left-tailed distributions result in positive and negative values, respectively. (B) Expression of genes located in the ori and ter proximal during overlapping cell cycles (1 to 3 h), transition (4 to 5 h) and non-overlapping cell cycles (6 to 10 h) growth phases. An asterisk indicates a significant difference in expression as inferred from a two-sided $t$ test with $P$-values corrected for multiple testing. (C) $\log_2$ FCs between time points from different growth phases. Red lines show the fitted general additive models; orange lines show the models shifted up with the conserved region at the terminus set to $\log_2$ FC of zero. Representative chromosome content indicative for the different growth phases is shown on the right.

The three components of the quorum sensing system showed different activation dynamics consistent with previous data (19, 20). While the primary QS activator *lasR/rsaL* pair was not among the significantly regulated genes, its relative expression showed a small but consistent gradually increase from 2 h cultivation on (Table S1). The *pqsABCDE*-operon was transiently activated with a peak between 4 and 5 h followed by a decline and therefore found in cluster 5. The pyocyanin biosynthesis gene clusters showed the same pattern, but with a much more pronounced peak. The chemotaxis operon was also activated transiently during in the transition phase (cluster 6), while flagella genes were not differentially regulated. The QS regulator RhlR was assigned to cluster 7 with an activation delay but stable expression throughout growth with non-overlapping cell cycles. The QS target genes coding for alkaline protease, cyanide production and lectin B were found in the same cluster.

**Influence of gene dosage on the transcriptome during overlapping cell cycles.** Next, we analyzed the distribution of genes in the determined clusters along the *ori-ter*-axis of the chromosome. Genes active during growth with overlapping cell cycles (clusters 1 to 3) were predominantly located close to *ori* while those that were activated during the transition phase (clusters 4 and 5) were more equally distributed along the chromosome. Genes in cluster 6, activated at the end of the transition phase, already showed a tendency toward *ter*, a trend that became even more pronounced for the genes in clusters 7 and 8 that increased expression during growth with non-overlapping cell cycles (Fig. 3A). Accordingly, the skewness of the position of genes relative to *ori* was positive for clusters 1 to 4 and negative for clusters 5 to 8, thus indicating a right and left tailed distribution, respectively. Furthermore, the average expression levels of genes in the *ori*-proximal half significantly exceeded those in the *ter*-proximal half of the chromosome

during the first 3 h of growth. At later time points, a balanced expression of both halves of the chromosome was observed (Fig. 3B). These data are in accordance with the predicted gene dosage effect in cells with high replication rates.

The presumed gene dosage effect became also visible when a general additive model was fitted to the $\log_2$ fold change transcriptome data along the chromosome in order to identify local trends in expression dynamics that go beyond the regulation of single genes or operons. When comparing subsequent time points, with a gradual change in chromosome content, a slightly lower expression was found around the terminus when transcriptomes from 3 h versus 4 h were compared (and to a lesser extent for 2 h versus 3 h; Fig. S3). This comparison marks the beginning of the transition from overlapping to non-overlapping cell cycles and also showed the strongest shift in chromosome content during cultivation (see Fig. 1C). However, these miniscule effects are probably negligible.

The position-specific differences in gene expression became more pronounced when we compared transcriptomes of time points with a higher difference in chromosome content (Fig. 3C). A clearly lower transcription of genes in the region surrounding the terminus of replication was visible when the different growth phases were compared, in particular seen for overlapping versus non-overlapping cell cycles. To a lesser extent this trend was also seen for the comparison of overlapping cell cycles to transition and transition to non-overlapping cell cycles. This pattern could in principle be the result of specific regulation events. However, in accordance with the analysis above, the reduction of gene expression proximal to, and also increasing toward *ter*, can be parsimoniously explained by a change in mRNA composition as a result of a higher transcriptional activity of *ori*-proximal genes, thus a gene-dosage effect (indicated by the orange line in Fig. 3C).

**Replication-associated transcriptome changes during non-overlapping cell cycles.** Growth with only one replication per cell division in the last 6 h of cultivation should allow to discriminate the transcriptomes of currently nonreplicating, replicating, and predivisional *P. aeruginosa* cells. To this end, we developed a protocol employing FACS to separate cells based on their chromosome content (Text S1). In order to determine the influence of fixation with formaldehyde (FA), and FACS on RNA composition, we compared samples obtained during different steps of the protocol to a sample fixed with RNAprotect (RP) (Fig. 4A). Across the three replicates, the different samples showed a consistently high correlation (Fig. 4B; Fig. S4A). We only found 15 genes as well as a chromosomal region of 32 phage-related genes, which were higher expressed in the RP- than in the FA-treated samples (Table S2). Only two genes found to be regulated during the cell cycle were also influenced by the fixation method, thus rendering the protocol suitable for the intended purpose.

Next, we compared the transcriptomes of the cell populations with one (C1) or two (C2) chromosomes and those replicating (R). The R and C2 fractions differed from the C1 fraction, but were highly similar to each other (Fig. 4C). Only 11 genes were found to be differentially expressed exclusively when these two fractions were compared. This included the *gnyDBHAL* gene cluster coding for enzymes of the acyclic isoprenoid degradation pathway (21), which showed the strongest downregulation in the R versus C2 fraction. The *nrdAB* genes coding for both subunits of the ribonucleotide-diphosphate reductase were downregulated in the C2 fraction compared to C1 and R. This enzyme catalyzes the last step in the formation of deoxyribonucleotides. In *E. coli*, it is activity has been linked to controlling the rate of DNA synthesis (22). Furthermore, it has been shown that gene expression peaks at initiation and declines toward the end of replication which is in accordance with our data for *P. aeruginosa*.

Between the actively replicating R and the C1 fraction, a clear dosage effect was visible with gene expression decreasing from *ori* to *ter* (Fig. 4D). The same was seen for the comparison of R and C2, but not when the fractions with only completely replicated chromosomes, C1 and C2 were compared (Fig. S4B). The differential expression of several chromosomal loci exceeded this trend dependent on the chromosomal position. In the R (and C2) fraction, the genes encoding the divisome showed the strongest activation compared to C1. These comprise of the *mur* and *mra* operons, encoding the

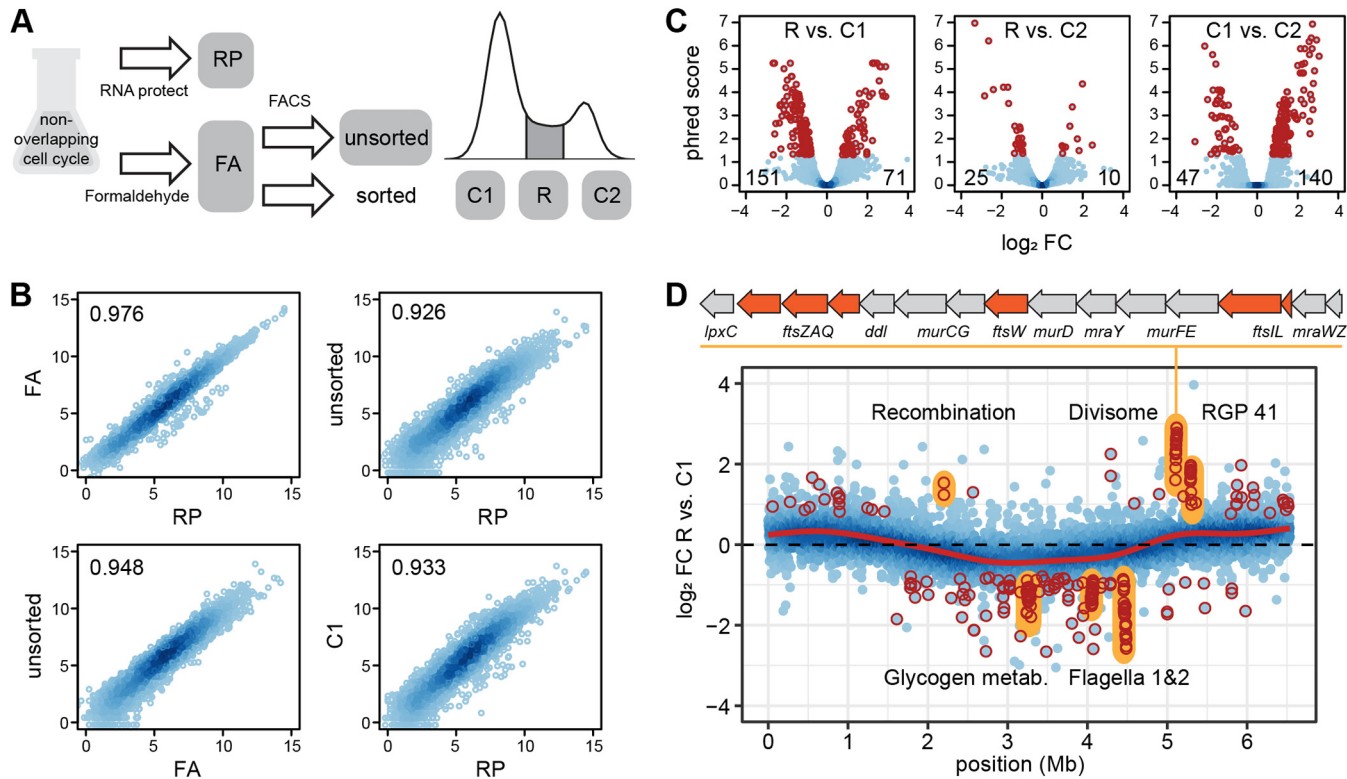

**FIG 4** Transcriptomes of replicating and nonreplicating cells during growth with non-overlapping cell cycles. (A) Sampling scheme for method evaluation. (B) Correlation between transcriptomes of differently treated RNAs. The axis indicate $\log_2$ counts per gene per million reads. Data for two additional replicates are shown in Fig. S4A. (C) Differential expression between replicating (R) and nonreplicating (C1, C2) cells. Number of significantly up- and downregulated genes between fractions (dark red) are shown in the left and right corner at the bottom of each panel, respectively. (D) Chromosome-wide differential gene expression in replicating (R) versus nonreplicating (C1) cells. Genes that change significantly in expression are marked in dark red. Operons discussed in the text are marked in yellow. The cell-division gene cluster is shown above the plot. The red line shows a fitted general additive model. Data for the comparisons R versus C2 and C2 versus C1 is shown in Fig. S4B.

enzymes for remodeling the peptidoglycan layer at the division plane and the *fts* genes, encoding the components responsible for septum formation (23). The recombination genes *lexA* and *recG* were upregulated, too. Of note was also the transcriptional activation of one genomic island, the region of genomic plasticity RGP41 (24), consisting of only uncharacterized genes. In the C1 fraction, the flagella gene clusters and chemotaxis operons, as well as the *glg* genes encoding the enzymes of the glycogen metabolic pathway showed the strongest activation compared to R and C2. Notably, the *mexE* gene, completely inactive in the other fractions, also showed a more than 64-fold higher expression in the C1 population, by far the strongest regulation in the data set (Table S2). It encodes the transmembrane protein part of an efflux-transporter for norfloxacin and imipenem (18).

## DISCUSSION

Here, we showed that *P. aeruginosa* switches from overlapping to non-overlapping cell cycles when cultivated in LB medium, thus allowing to study the effect of replication on the transcriptome. Here, the chromosomal gene order reflects the expression maxima during both growth-phases with the genes important for fast growth being located closer to *ori* and the stationary-phase genes located closer to *ter*. It has been demonstrated before that the *E. coli* sigma 70 factor and its targets, which are mostly active in the exponential phase, are located closer to *ori*, while the sigma S factor and its mostly stationary-phase active targets are located closer to *ter* (25). Thus, while the sigma factors transcriptionally regulate downstream genes, regulon expression is additionally enhanced by a gene dosage effect that can act on the regulators and their target genes. Our data show the potential of combining identification of different growth

phases by flow cytometry with the comparison of the respective transcriptomes. The gained knowledge could generally be used to identify replication-associated effects on gene expression for the vast number of strains with existing transcriptome data (14, 16, 26), and integrated into existing gene regulatory models (15, 27). It could further help to better understand chromosomal architecture and to explain gene order evolution (2, 25, 28, 29).

When growing with non-overlapping cell cycles, *P. aeruginosa* displays a distinct transcriptome between the approximately 25% currently dividing and 75% currently nondividing cells. Expression of flagella genes is restricted to cells that are not replicating, while those that replicate differ mainly in the activity of a cell division locus. Furthermore, we found that expression of *mexE*, part of an important antibiotic resistance trait, is restricted to the nondividing cells. Two different scenarios could explain the distribution of nonreplicating and replicating cells: The cells in the C1 fraction could either be in a prolonged phase of preparation for the next round of replication or they could be in a growth-arrested state; thus, the population would be divided into actively dividing and nondividing cells. If the latter case is true, induction of subpopulations during the switch in growth phases would be coincidental with the activation of the *rhl* QS system. Cell communication induced population heterogeneity has been shown for *P. aeruginosa* before (30) and is also common in other bacteria (31–33). It might also be the trigger switching the replication mode and restricting activity of the flagella gene clusters to the nondividing cells. In contrast to chemotaxis, flagella gene expression has not been described to be controlled directly by QS before (19, 20). However, we also did not find them differentially expressed in the culture as a whole, but only in a subpopulation. Thus, a possible connection between communication and development of motility in a fraction of cells might have been overlooked and is worth a closer investigation. Furthermore, slow-growing QS-defective mutants frequently evolve during CF infections (34, 35) It would be interesting to determine if these strains reproduce by non-overlapping cell cycles only and how the transcriptome is affected by this change.

The highly similar transcriptomes of actively replicating and predivisional cells indicate that, in *P. aeruginosa*, no distinct phases of a differentiation program are coupled to progressing replication. This is in stark contrast to the precisely timed cell cycle of *C. crescentus* with a defined order of gene activity as cells replicate (36). Transcriptome dynamics during replication has so far only been determined for a couple of model bacteria (36–39). Key to these studies was the ability to synchronize the cell cycle within the cultures. Our newly developed method based on cell sorting according to DNA content allows for identification of replication-specific gene expression without the need for synchronization, as long as the cells grow slowly with one round of replication per cell division. Not only cell sorting, but also complementary recent advances in single cell sequencing (40), open up the path to comparative analysis of larger groups of bacteria, thus contributing to a better understanding of the evolution of cell cycle control at the transcriptional level (41).

## MATERIALS AND METHODS

**Strains and growth conditions.** *Pseudomonas aeruginosa* PA14 (42) was grown in Lysogeny Broth (10 g/L tryptone, 5 g/L yeast extract, 10 g/L NaCl) at 37°C and 160 rpm shaking. The growth of cultures inoculated from an overnight culture to a starting $OD_{600}$ of 0.05 was followed for 10 h and samples for determination of $OD_{600}$, cell count, DNA content, and RNAseq were withdrawn every hour. For FACS-based sorting, cultures were inoculated to a starting $OD_{600}$ of 0.2 and samples were prepared after 5 h when the growth mode with non-overlapping cell cycles was stably reached.

**Flow cytometric determination of cell number and chromosome content.** A total of 100 $\mu$L of culture were mixed with 80 $\mu$L of 25% glutaraldehyde in H2O and incubated for 5 min. Then, 820 $\mu$L of PBS were added and a dilution series up to 1:1,000 was prepared; 10 $\mu$L of SYBR green (100×) was added to 1 mL of fixed and diluted sample. After an incubation time of 20 min, the sample was measured on a BD FACS Canto flow cytometer (BD Biosciences, Heidelberg, Germany). After gating based on centered forward and sideward scatter, cells were identified and chromosome content quantified by fluorescence detection in the FITC channel (excitation 488 and emission 535 nm). Data processing and analysis were performed with the R package ggcyto (43).

**Electron microscopy.** Bacteria were fixed by addition of glutaraldehyde (final concentration 2%) for 30 min, and addition of formaldehyde (final concentration 5%) into the culture medium. EM sample preparation was performed as previously described (44) with slight modifications. In brief, samples were washed twice with TE-buffer and fixed to poly-l-lysine coated round coverslips. After additional washing steps, the samples were dehydrated in a gradient series of acetone (10%, 30%, 50%, 70%, 90%) on ice and two steps with 100% acetone at room temperature (each step for 10 min). Afterwards, samples were critically point dried with the CPD300 (Leica Microsystems, Wetzlar, Germany), mounted to aluminum pads and sputter coated with gold-palladium. Images were acquired with a field emission scanning electron microscope Merlin (Zeiss, Jena, Germany) equipped with an Everhart Thornley and an inlens detector and operating at an acceleration voltage of 5 kV.

**RNAseq library preparation from whole cultures.** Depending on the density, 1 to 2 mL of culture were mixed with the same volume RNAprotect Bacteria Reagent (Qiagen, Hilden, Germany) incubated for 10 min and centrifuged. The pellets were flash-frozen and stored at −70°C. RNA extraction was carried out with the RNeasy Plus Kit in combination with QIAshredder columns (Qiagen, Hilden, Germany). Treatment with DNase I was performed in solution. Multiplexed libraries were generated from directly barcoded fragmented RNA according to a previously published custom protocol (45), including rRNA removal with the RiboZero Kit (Illumina, San Diego, USA).

**Fluorescence-activated cell sorting for RNAseq of subpopulations.** The method was developed based on a previously published study (46). A step-by-step protocol for sample preparation, sorting, and RNA isolation is provided in Text S1. Key to successful RNA recovery is the gentle formaldehyde fixation at 4°C. Aliquots of fixed samples were adjusted to approximately $1.8 \times 10^7$ cells/mL in 30 mL volume each and stained with SYBR green. Sorting of $5.4 \times 10^8$ cells based on the FITC-signal (see above) directly into RNAprotect was performed with the BD FACSAria Fusion (BD Biosciences, Heidelberg, Germany). The sorted cells were collected on a filter from which RNA was extracted using a combination of Lysozyme and Proteinase K digestion with bead beating, and purified with NucleoZOL (TaKaRa Bio, Göteborg, Sweden). rRNA depletion was performed with the NEBNext Bacteria kit (NEB, Frankfurt, Germany). The libraries were prepared with the TruSeq kit (Illumina, San Diego USA).

**Transcriptome analysis.** Sequencing of all libraries was performed on a NovaSeq 6000 (Illumina, San Diego, USA) in paired-end mode with 100 cycles in total. Reads were filtered with fastQC-mcf (https://github.com/ExpressionAnalysis/ea-utils) and mapped to the *P. aeruginosa* PA14 genome (RefSeq accession GCF_000404265.1) using bowtie2 (47). FeatureCounts was used to assess the number of reads per gene (48). Normalization and identification of significantly differentially regulated genes (FDR $<$ 0.05, absolute log$_2$ fold change [FC] $>$ 1) was performed in R using the glmTreat-function of edgeR (49) with correction for false discovery rate using the method by Benjamini and Hochberg (50). Cluster assignment of differentially expressed genes was performed with the package mfuzz (51). Metabolic pathway annotation was obtained from the Kyoto encyclopedia of genes and genomes. Identification of significantly enriched KEGG pathways in the clusters was performed using the hypergeometric test (phyper in the R stats package). The obtained *P*-values were corrected for false discovery rate using the method by Benjamini and Hochberg (50). To test whether the gene positions in the different clusters are predominantly located closer to *ori* or *ter*, we calculated their distance to *ori* and determined the skewness of the relative position using the respective function in the R package e1071. Positive and negative value indicate a tailed to the right and left, respectively. To test for significant differences in the expression of genes closer to *ori* or *ter*, we divided the chromosome into two halves. A *t* test was employed to compare the means of expression on both halves. The obtained *P*-values from 10 different comparisons were corrected for false discovery rate. The smoothing line for log$_2$ FCs along the chromosome axis was constructed by employing a general additive model with cubic splines using the geom_smooth function with formula y $\sim$ s(x, bs = "cs") in the R package ggplot2.

**Data availability.** RNAseq raw data have been deposited at the NCBI gene expression omnibus database under accessions GSE159698 and GSE217100 .

## SUPPLEMENTAL MATERIAL

Supplemental material is available online only.

**TEXT S1**, PDF file, 0.7 MB.
**FIG S1**, TIF file, 1 MB.
**FIG S2**, TIF file, 2.3 MB.
**FIG S3**, TIF file, 2.2 MB.
**FIG S4**, TIF file, 2.2 MB.
**TABLE S1**, XLSX file, 1.5 MB.
**TABLE S2**, XLSX file, 1.2 MB.

## ACKNOWLEDGMENTS

S.H. was funded by the Lower Saxony Ministry for Science and Culture (Bacdata ZN3428), the European Union (EU, ERC Consolidator Grant COMBAT 724290) and by the Deutsche Forschungsgemeinschaft (DFG, German Research Foundation) under Germany's Excellence Strategy (EXC 2155 390874280). Furthermore, S.H. received additional funding from the DFG (DFG SPP 1879) and the Novo Nordisk Foundation (NNF 18OC0033946). J.T.

received funding from the mobility grant of the Czech Ministry of Education (CZ.02.2.69/0.0/0.0/18_053/0017705). We are grateful to Astrid Dröge and Ina Schleicher for technical support. We thank the reviewers for their critical and constructive assessment of our manuscript.

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
