## [Reviewer comments · mSystems]

Transcriptome dynamics of *Pseudomonas aeruginosa* during transition from overlapping to non-overlapping cell cycles

Katrin Alpers, Elisabeth Vatareck, Lothar Gröbe, Mathias Müssen, Maren Scharfe, Susanne Häussler, and Jürgen Tomasch

Corresponding Author(s): Jürgen Tomasch, Institute of Microbiology, CAS Centre Algatech

Review Timeline:

Submission Date:	November 16, 2022
Editorial Decision:	January 3, 2023
Revision Received:	January 19, 2023
Accepted:	January 22, 2023

Editor: Oleg Igoshin

Reviewer(s): The reviewers have opted to remain anonymous.

Transaction Report:

DOI: <https://doi.org/10.1128/msystems.01130-22>

January 3, 2023

Dr. Jürgen Tomasch
Institute of Microbiology, CAS Centre Algatech
Novohradská 237
Třeboň 37901
Czech Republic

Re: mSystems01130-22 (Transcriptome dynamics of *Pseudomonas aeruginosa* during transition from replication-uncoupled to -coupled growth)

Dear Dr. Jürgen Tomasch:

Thank you for submitting your manuscript to mSystems. We have completed our review and I am pleased to inform you that, in principle, we expect to accept it for publication in mSystems. However, acceptance will not be final until you have adequately addressed the reviewer comments.

Preparing Revision Guidelines

Sincerely,

Oleg Igoshin

Editor, mSystems

Journals Department
American Society for Microbiology

Reviewer comments:

Reviewer #1 (Comments for the Author):

Alpers et al. study replication-phase dependent global transcription in *Pseudomonas aeruginosa*. They capture transcriptomes on an hourly basis in growing PAE cultures, while measuring chromosome content and cell sizes, which allows them to differentiate growth phase where replication is uncoupled to division and where it is coupled. In addition, they separate (fixed) cells by FACS on the basis of their chromosome content and re-analyze transcriptome differences. Their data suggest growth phase and cell-cycle dependent transcriptional clusters, and an on average higher expression of genes close to the chromosomal origin than terminus of replication.

COMMENTS

- 1) The manuscript is relatively straightforward and easy to follow. The data are well represented and mostly properly analyzed as far as I can judge. The main message is sort of known from other bacterial strains, but has so far not been quantified for *Pseudomonas aeruginosa*.
- 2) I have a few specific context questions and some suggestions for other or different data presentation, as specified below.
- 3) l. 55 is the 'speed of DNA replication' a constant (per species), or is it dependent on the availability of nutrients? You seem to present it here as a given, but your results suggest it is not. Maybe be more differentiated here?
- 4) l. 173-177: How can you design a proper statistical test to underpin your statement that the distribution of the 'active' genes along the chromosome is (i) different and (ii) closer to the ori vicinity for phase 1-3 and more to the ter vicinity for phase 5-8? see Figure 3A.
- 5) What is the 'general additive model' doing (L. 185)? Where do you describe it? The effect in Fig S3 is maximally one-fifth of a log₂ value... Is that even significant? Figure 3C
- 6) In relation to that: for your read mapping, do you take into consideration that genes close to the ori would be partly duplicated? Is the smiley effect you would get from DNA sequencing the same as from the mapped log FC Rvs C1 transcriptome?
- 7) l. 192 Not sure what means 'strongest shift in chromosome content' (which content do you mean?)
- 8) l. 254-260: so how does this explanation compare to *P. aeruginosa*, then? Is it the gene dosage on the sigma factors or the gene copies per se that would cause higher expression of ori-vicinity genes? In this argument, do you take into consideration that sigma S expression is more translationally regulated than transcriptionally?
- 9) l. 262-266 Can you give examples here for these suggestions?
- 10) l. 268-269: I don't understand this statement, and I could imagine discussing this a bit more in detail. I presumed that individual cells in the phase 5-8 are going alternately through a replication state and a non-replicating state, rather than having 25% of all cells in the population always replicating and 75% not replicating at all. Are you suggesting that individual cells would 'cycle' in gene expression (as in Fig. 4D) as they go from a replicating stage to a dividing stage? In that case, would you not expect a carry-over of expressed genes from one phase into the next? (Or perhaps differentially depending on the gene).
- 11) Fig 2B: what means 'ratio enriched vs all'? Many KEGG categories here have values below 0.2; does that mean there are fewer than 20% of the total number of genes in this KEGG category that are differentially expressed in that cluster? How much would you expect by chance?
- 12) Fig. 3A what are genes that show the 'highest expression'? What is the criterium here? How can you conclude from this if there is really a gene dosage effect on transcription? (l. 182)
- 13) Fig. 3B can you test whether the expression distributions between ori and ter are statistically significantly different? Is your division here between ori and ter region justified? For example, Fig. 3C seems to suggest that the 'ori' region would be between chromosome 0-2 and 4-6 Mb, and ter 'only' between 2-4 Mb.
- 14) Fig. 3C How can you exclude that there is not a specific gene silencing effect for ter-vicinity genes? (Rather than a specific up-regulation for ori-vicinity genes, or consequences of gene dosage)?

Reviewer #2 (Comments for the Author):

Overall the work by Alpers et al. is an interesting investigation into the role of replication and growth state in determining the transcriptome of *Pseudomonas aeruginosa*, a major human pathogen. While studies of this sort have been conducted chiefly in model organisms such as *E. coli*, how broadly these conclusions hold is not really clear, meaning that expanding to important non-model organisms is a significant contribution from this paper. The major innovation in this work is using a FACS-based approach to separate cells based on cell cycle state for transcriptome analysis. The authors take care in validating this approach, so that not only will these datasets be useful for *P. aeruginosa* researchers but the techniques involved may be applied to other species too.

Major comments:

While overall I believe this study is well-designed and well-executed, I do have concerns around the conceptualization of "coupled" and "uncoupled" growth:

1. The classification of "coupled" growth states as non-dividing, replicating, and pre-divisional is perhaps not well-supported. In *E. coli*, the terminology of "B, C, and D" periods in slow-growing cells roughly corresponds to these states. However, applying the term "non-dividing" to cells with one chromosome copy implies they are not actively cycling, whereas classical studies on synchronized *E. coli* would support that they may just be undergoing sufficient growth in order to initiate DNA replication. It is hard to tell here whether all "non-dividing" cells are undergoing this pre-replication growth or whether there is a mixture of truly non-growing and "B-period" cells. This is especially true as neither the environment nor the growth rate is likely to be constant over time under these conditions. For this reason, I find it hard to justify the inference that 25% of cells are replicating and of these their doubling time is 42 min (lines 120-121). We simply can't tell how many cells are undergoing their cell cycle from these data, and hence we can't infer the doubling time of those that are.
2. The authors reduce changes in the transcriptome to this binary transition from "coupled" to "uncoupled" growth. However, a large effect happening concurrent with this is environmental changes (exhaustion of carbon and other nutrient sources, accumulation of waste, changes in pH etc.) This is not really discussed and given that the authors extensively describe enrichments within gene expression patterns during growth, more consideration of these factors is warranted.
3. I would dispute the term "uncoupled" growth. Classical studies in *E. coli* show that under fast growth conditions, replication is very tightly coupled to division, albeit with a different pattern depending on growth rate. There's no reason to believe that this would be any different in *P. aeruginosa*.

Minor comments:

1. I think when comparing growth phases the authors should be consistently clear that they are describing relative abundance of gene transcripts. This is important since for example on line 178-179 they say that cluster 7 and 8 genes showed increased expression during coupled growth. However, in reality it is feasible that overall transcriptional activity at these late growth stages may be well down, and so this might represent less of an induction of these genes and more of a loss of the others. The authors do touch on this idea (e.g. orange line in Figure 3C) but it could be more visible.
2. More detail could be given on the general additive model.
3. For correlation plots (e.g. Figure 4B) I assume the axes represent log-expression but it's not stated as far as I can see.
4. The supplementary protocol for the fixation and isolation is a great addition that will help increase the impact of this work. However, please could the authors define the reagent "ATL" used during crosslinking reversal.

Reviewer #1 (Comments for the Author):

Alpers et al. study replication-phase dependent global transcription in *Pseudomonas aeruginosa*. They capture transcriptomes on an hourly basis in growing PAE cultures, while measuring chromosome content and cell sizes, which allows them to differentiate growth phase where replication is uncoupled to division and where it is coupled. In addition, they separate (fixed) cells by FACS on the basis of their chromosome content and re-analyze transcriptome differences. Their data suggest growth phase and cell-cycle dependent transcriptional clusters, and an on average higher expression of genes close to the chromosomal origin than terminus of replication.

COMMENTS

1) The manuscript is relatively straightforward and easy to follow. The data are well represented and mostly properly analyzed as far as I can judge. The main message is sort of known from other bacterial strains, but has so far not been quantified for *Pseudomonas aeruginosa*.

2) I have a few specific context questions and some suggestions for other or different data presentation, as specified below.

Answer: Thank you! We appreciate your valuable comments and suggestions.

3) I. 55 is the 'speed of DNA replication' a constant (per species), or is it dependent on the availability of nutrients? You seem to present it here as a given, but your results suggest it is not. Maybe be more differentiated here?

Answer: We are sorry for the imprecise wording here. As suggested by the other reviewer we replaced the terms coupled and uncoupled growth by overlapping and non-overlapping cell cycles and in course of rewriting we also rephrased this sentence. What we want to point out here is that fast growing bacteria initiate a new round of replication before the previous one is finished resulting in overlapping cell cycles. The term "speed" does not really express this. We hope the modified introduction is clearer now.

4) I. 173-177: How can you design a proper statistical test to underpin your statement that the distribution of the 'active' genes along the chromosome is (i) different and (ii) closer to the ori vicinity for phase 1-3 and more to the ter vicinity for phase 5-8? see Figure 3A.

Answer: Thank you for this question. We decided to test for skewness of the gene positions in relation to the distance to the origin of replication in the different clusters. If more genes are closer to ori we would expect a right tailed distribution and if more genes are closer to ter, we would expect a left tailed distribution of the relative positions. This is exactly what we see in the skewness values that are positive (right tailed) for clusters 1 to 4 and negative (left tailed) for clusters 5 to 8. The highest skews positive and negative skews are observed for cluster 3 and 7, respectively. We added the skewness values to the figure and an explanation to the figure legend.

5) What is the 'general additive model' doing (L. 185)? Where do you describe it? The effect in Fig S3 is maximally one-fifth of a log₂ value... Is that even significant? Figure 3C

Answer: We are sorry for the missing information. The following sentence has been added to the material and methods section: "The smoothing line for log₂ FCs along the chromosome axis was constructed by employing a general additive model with cubic splines using the `geom_smooth` function with formula $y \sim s(x, bs = "cs")$ in `ggplot2`". We agree that the observed position-dependent effect on transcription is rather small when subsequent time points are compared. We also think that these effects are rather negligible and added this estimate to the text.

6) In relation to that: for your read mapping, do you take into consideration that genes close to

the ori would be partly duplicated? Is the smiley effect you would get from DNA sequencing the same as from the mapped log FC Rvs C1 transcriptome?

Answer: We did not sequence DNA from the different time points, although in hindsight, this would have been a valuable addition of data. RNAseq reads were mapped on the genome sequence that does not contain coverage information. From the estimation of chromosome content by flow cytometry we conclude that during the first time points, the cells grow with overlapping cell cycles which would result in the “smiley effect” when DNA would have been sequenced. The observed “smiley effect” in the transcriptome data would be a direct consequence of the gene dosage effect that is a result of the actively replicating chromosome.

7) I. 192 Not sure what means 'strongest shift in chromosome content' (which content do you mean?)

Answer: We are now referring to Figure 1C after this sentence to make clear what we mean. Between 3 and 4 h of growth the chromosome content of the cells shifts back from more than 3 chromosome equivalents to a maximum of 2. This is nicely visible from the density plots of the SYBR Green staining.

8) I. 254-260: so how does this explanation compare to *P. aeruginosa*, then? Is it the gene dosage on the sigma factors or the gene copies per se that would cause higher expression of ori-vicinity genes? In this argument, do you take into consideration that sigma S expression is more translationally regulated than transcriptionally?

Answer: We cited this reference as it is in our opinion a good example of the integration of genome structure and gene regulation. It is meant more as a suggestion for further analysis than as an exhaustive discussion. In future we plan to integrate the generated transcriptomes with existing data on regulons and also perform proteome experiments. However, this will require further substantial effort.

9) I. 262-266 Can you give examples here for these suggestions?

Answer: Our intention here is to encourage other researchers to take positional data, information on the growth stage and replicative state as we did in the current manuscript into consideration when analysing their own data or designing new experiments.

10. I. 268-269: I don't understand this statement, and I could imagine discussing this a bit more in detail. I presumed that individual cells in the phase 5-8 are going alternately through a replication state and a non-replicating state, rather than having 25% of all cells in the population always replicating and 75% not replicating at all. Are you suggesting that individual cells would 'cycle' in gene expression (as in Fig. 4D) as they go from a replicating stage to a dividing stage? In that case, would you not expect a carry-over of expressed genes from one phase into the next? (Or perhaps differentially depending on the gene).

Answer: Thank you for pointing to this weakness in our discussion. Indeed, two different scenarios can explain the observed distribution of chromosome content during the later growth stages. The following sentences have been added to the discussion: “Two different scenarios could explain the distribution of cells in non-replicating and replicating populations: The cells in the C1 fraction could either be in a prolonged phase of preparation for the next round of replication or they could be in a growth-arrested state, thus, the population would differentiate into actively dividing and non-dividing cells. If the latter case is true, induction of subpopulations during the switch in growth phases would be coincidental with the activation of the rhl QS system.”

11) Fig 2B: what means 'ratio enriched vs all'? Many KEGG categories here have values below

0.2; does that mean there are fewer than 20% of the total number of genes in this KEGG category that are differentially expressed in that cluster? How much would you expect by chance?

Answer: We apologize for not giving the details on the KEGG category enrichment analysis in the materials and methods section and for the missing details in the figure legend. This has been done now in the revised manuscript. We added the following to the method section: "Metabolic pathway annotation was obtained from the Kyoto encyclopaedia of genes and genomes. Identification of significantly enriched KEGG pathways in the clusters was performed using the hypergeometric test (phyper in the base R package). The obtained p-values were corrected for false discovery rate using the method by Benjamini and Hochberg." As stated, the plot only shows KEGG pathways that are significantly enriched in the clusters (fdr-corrected p-value < 0.05). Given the small size of the clusters compared to the rather large genome of *P. aeruginosa*, an enrichment of less than 20% can also be significant.

12) Fig. 3A what are genes that show the 'highest expression'? What is the criterium here? How can you conclude from this if there is really a gene dosage effect on transcription? (l. 182)

Answer: We refer here to the identified clusters and their expression profiles as shown in Figure 2A. We added this information to the Figure legend. We want to point out that every single analysis is only indicative of a gene dosage effect. We conclude from all three analysis described here that we can observe a gene dosage effect in our data.

13) Fig. 3B can you test whether the expression distributions between ori and ter are statistically significantly different?

Is your division here between ori and ter region justified? For example, Fig. 3C seems to suggest that the 'ori' region would be between chromosome 0-2 and 4-6 Mb, and ter 'only' between 2-4 Mb.

Answer: We sought to test here if genes on the chromosome closer to ori show on average a different expression level from genes closer to the terminus. Therefore, we divided them into two according halves. The advantage is an almost equal sample size of genes in the two datasets. As the expression values are also close to a normal distribution we now employed a two-sided t-test to test if the means are significantly different for each time point. We corrected the obtained p-values for multiple testing, as we analysed ten datasets. The result shows that there is a significant difference for the first three hours. This information has been added to the figure and figure legend as well as the material and methods section.

14) Fig. 3C How can you exclude that there is not a specific gene silencing effect for ter-proximity genes? (Rather than a specific up-regulation for ori-proximity genes, or consequences of gene dosage)?

Answer: In principle a specific up- or down-regulation of ori- or respectively ter-proximal could also explain the observed pattern. However, we think this scenario is unlikely as we can clearly see a gradual decrease in the fold change data towards ter that can be better explained by position specific effects due to gene dosage differences. The other analysis in this section as well as the similar pattern in the sorted cells supports in our opinion this interpretation. In particular in the transcriptomes of sorted cells, positional effects and specific regulation are clearly distinguishable. Nevertheless, we added this alternative explanation to the text: "This pattern could in principle be the result of specific regulation events. However, in accordance with the analysis above, the reduction of gene expression proximal to, and also increasing towards ter, can be parsimoniously explained by a change in mRNA composition as a result of a higher transcriptional activity of ori-proximal genes, thus a gene-dosage effect (indicated by the orange line in Figure 3C)."

Reviewer #2 (Comments for the Author):

Overall the work by Alpers et al. is an interesting investigation into the role of replication and growth state in determining the transcriptome of *Pseudomonas aeruginosa*, a major human pathogen. While studies of this sort have been conducted chiefly in model organisms such as *E. coli*, how broadly these conclusions hold is not really clear, meaning that expanding to important non-model organisms is a significant contribution from this paper. The major innovation in this work is using a FACS-based approach to separate cells based on cell cycle state for transcriptome analysis. The authors take care in validating this approach, so that not only will these datasets be useful for *P. aeruginosa* researchers but the techniques involved may be applied to other species too.

Answer: Thank you! We appreciate your valuable comments and suggestions.

Major comments:

While overall I believe this study is well-designed and well-executed, I do have concerns around the conceptualization of "coupled" and "uncoupled" growth:

1. The classification of "coupled" growth states as non-dividing, replicating, and pre-divisional is perhaps not well-supported. In *E. coli*, the terminology of "B, C, and D" periods in slow-growing cells roughly corresponds to these states. However, applying the term "non-dividing" to cells with one chromosome copy implies they are not actively cycling, whereas classical studies on synchronized *E. coli* would support that they may just be undergoing sufficient growth in order to initiate DNA replication. It is hard to tell here whether all "non-dividing" cells are undergoing this pre-replication growth or whether there is a mixture of truly non-growing and "B-period" cells. This is especially true as neither the environment nor the growth rate is likely to be constant over time under these conditions. For this reason, I find it hard to justify the inference that 25% of cells are replicating and of these their doubling time is 42 min (lines 120-121). We simply can't tell how many cells are undergoing their cell cycle from these data, and hence we can't infer the doubling time of those that are.

Answer: We agree that the chosen terminology is unfortunate. We replaced the term uncoupled growth by growth with overlapping cell cycles and the term coupled growth by non-overlapping cell cycles and modified the text accordingly. These terms, used by Reyes-Lamothe and Sheratt in their recent review of the bacterial cell cycle are more precise and hopefully better understandable. The other reviewer also rightfully criticized that we talked about different speeds of DNA replication. We now made clear that the difference between overlapping and non-overlapping cell cycles is that a new round of replication is started before the previous one is finished. We deleted the last sentence about estimating replication times (lines 120-121). We agree that a robust estimate would require e.g. time-lapse microscopic observations that we did not perform here. We also discuss now the two options that could explain the observed flow cytometry pattern: "Two different scenarios could explain the distribution of non-replicating and replicating cells: The cells in the C1 fraction could either be in a prolonged phase of preparation for the next round of replication or they could be in a growth-arrested state, thus, the population would be divided into actively dividing and non-dividing cells. If the latter case is true, induction of subpopulations during the switch in growth phases would be coincidental with the activation of the rhl QS system."

2. The authors reduce changes in the transcriptome to this binary transition from "coupled" to "uncoupled" growth. However, a large effect happening concurrent with this is environmental changes (exhaustion of carbon and other nutrient sources, accumulation of waste, changes in pH etc.) This is not really discussed and given that the authors extensively describe enrichments within gene expression patterns during growth, more consideration of these factors is warranted.

Answer: Thank you for this comment. We agree that the generated data can be used in a much broader way than we did here. To our knowledge we performed the first transcriptome analysis following the growth of *P. aeruginosa* since Schuster et al. (2003) who back then used microarrays. Thus, we expect that the dataset, that we made publically available, is of value as a reference for the whole *Pseudomonas* community. Furthermore, we are currently working on another study that focuses on the metabolic dynamics of *P. aeruginosa* during growth that will also employ the generated transcriptome dataset in combination with other experiments. The current manuscript focusses on the identification of gene dosage effects and we had the feeling that an exhaustive discussion of all the interesting transcriptional changes that we observed and also briefly describe in the results section (including some thoughts about nutrient availability) would blur the message that we intend to deliver to the reader. For these reasons we would rather keep the discussion restricted to the potential influence of cell cycle dynamics on the transcriptome.

3. I would dispute the term "uncoupled" growth. Classical studies in *E. coli* show that under fast growth conditions, replication is very tightly coupled to division, albeit with a different pattern depending on growth rate. There's no reason to believe that this would be any different in *P. aeruginosa*.

Answer: We agree and removed the term uncoupled growth from the manuscript. See the detailed response to point 1 for further explanation.

Minor comments:

1. I think when comparing growth phases the authors should be consistently clear that they are describing relative abundance of gene transcripts. This is important since for example on line 178-179 they say that cluster 7 and 8 genes showed increased expression during coupled growth. However, in reality it is feasible that overall transcriptional activity at these late growth stages may be well down, and so this might represent less of an induction of these genes and more of a loss of the others. The authors do touch on this idea (e.g. orange line in Figure 3C) but it could be more visible.

Answer: We fully agree on this point. Indeed, what you get with RNAseq data is the composition of RNA in the cell. If the abundance of the majority of RNA molecules does not change, it is justified to speak of activation or repression. However, in our case, when cells undergo reductive growth during the transition to non-overlapping cell cycles, major changes in the overall composition of the RNA pool are expected. Therefore, in the revised manuscript we now avoid terms like activation and made clear that we talk about relative expression when describing changes in the transcriptome. However, we made an exception for the quorum sensing system. Its activation dynamics has been well characterized in numerous publications and with different techniques therefore we think it is justified to use the term activation in this case.

2. More detail could be given on the general additive model.

Answer: We are sorry for the missing information. The following sentence has been added to the material and methods section: "The smoothing line for log₂ FCs along the chromosome axis was constructed by employing a general additive model with cubic splines using the `geom_smooth` function with formula $y \sim s(x, bs = "cs")$ in `ggplot2`."

3. For correlation plots (e.g. Figure 4B) I assume the axes represent log-expression but it's not stated as far as I can see.

Answer: The missing information has been added to both figure legends. "The axis indicate log₂ counts per gene per million reads."

4. The supplementary protocol for the fixation and isolation is a great addition that will help increase the impact of this work. However, please could the authors define the reagent "ATL" used during crosslinking reversal.

Answer: We are sorry for the missing information that has now been added. We used the ATL buffer and Proteinase K solution from excess QIAGEN DNeasy blood & tissue in our lab, however both can be also ordered separately. This is now clearly stated in the protocol.

January 22, 2023

Dr. Jürgen Tomasch
Institute of Microbiology, CAS Centre Algatech
Novohradská 237
Třeboň 37901
Czech Republic

Re: mSystems01130-22R1 (Transcriptome dynamics of *Pseudomonas aeruginosa* during transition from overlapping to non-overlapping cell cycles)

Dear Dr. Jürgen Tomasch:

Your manuscript has been accepted, and I am forwarding it to the ASM Journals Department for publication. For your reference, ASM Journals' address is given below. Before it can be scheduled for publication, your manuscript will be checked by the mSystems production staff to make sure that all elements meet the technical requirements for publication. They will contact you if anything needs to be revised before copyediting and production can begin. Otherwise, you will be notified when your proofs are ready to be viewed.

If you would like to submit a potential Featured Image, please email a file and a short legend to msystems@asmusa.org. Please note that we can only consider images that (i) the authors created or own and (ii) have not been previously published. By submitting, you agree that the image can be used under the same terms as the published article. File requirements: square dimensions (4" x 4"), 300 dpi resolution, RGB colorspace, TIF file format.

We recognize that the video files can become quite large, and so to avoid quality loss ASM suggests sending the video file via <https://www.wetransfer.com/>. When you have a final version of the video and the still ready to share, please send it to mSystems staff at msystems@asmusa.org.

Sincerely,

Oleg Igoshin
Editor, mSystems

Journals Department
E-mail: mSystems@asmusa.org